# IL-31: State of the Art for an Inflammation-Oriented Interleukin

**DOI:** 10.3390/ijms23126507

**Published:** 2022-06-10

**Authors:** Francesco Borgia, Paolo Custurone, Federica Li Pomi, Raffaele Cordiano, Clara Alessandrello, Sebastiano Gangemi

**Affiliations:** 1Department of Clinical and Experimental Medicine, Section of Dermatology, University of Messina, Messina, Italy C/O A.O.U.P. “Gaetano Martino”, Via Consolare Valeria 1, 98125 Messina, Italy; paolo.custurone@gmail.com (P.C.); federicalipomi@hotmail.it (F.L.P.); 2Department of Clinical and Experimental Medicine, School and Operative Unit of Allergy and Clinical Immunology, University of Messina, 98125 Messina, Italy; raffaelecordiano@gmail.com (R.C.); clara.alessandrello@outlook.it (C.A.); sebastiano.gangemi@unime.it (S.G.)

**Keywords:** IL-31, skin diseases, hematological diseases, cancer, biological drugs, respiratory diseases, allergic diseases, interleukin-31, itch, interleukins

## Abstract

Interleukin 31 belongs to the IL-6 superfamily, and it is an itch mediator already studied in several diseases, comprising atopic dermatitis, allergic pathologies, and onco-hematological conditions. This research aims to assess the role of this cytokine in the pathogenesis of these conditions and its potential therapeutic role. The research has been conducted on articles, excluding reviews and meta-analysis, both on animals and humans. The results showed that IL-31 plays a crucial role in the pathogenesis of systemic skin manifestations, prognosis, and itch severity. Traditional therapies target this interleukin indirectly, but monoclonal antibodies (Mab) directed against it have shown efficacy and safety profiles comparable with biological drugs that are already available. Future perspectives could include the development of new antibodies against IL-31 both for humans and animals, thus adding a new approach to the therapy, which often has proven to be prolonged and specific for each patient.

## 1. Introduction

### 1.1. IL-31 Structure

Interleukin 31 (IL-31) belongs to the IL-6 superfamily, which comprises other members, such as IL-11, IL-21, IL-27, oncostatin M (OSM), leukemia inhibitory factor (LIF), and cardiotrophin 1 (CT-1). Like other IL-6 family cytokines, IL-31 is an anti-parallel four-helix bundle, with an “up-up-down-down” topology in short chains, with an early form composed of 164 amino acids (aa) and a mature form of 141 aa [1].

### 1.2. IL-31 Release Mechanism

This cytokine is produced constitutively by immune cells, such as CD4+ Th2 cells, monocyte/macrophages, and dendritic cells [2], but is also expressed in some non-immune cells, such as fibroblasts and keratinocytes [3]. In particular, a subpopulation of CD4+ T cells (CD45RO+) mainly produces IL-31. CD45RO+ T cells have a very distinct epidermotropism, meaning they migrate selectively into the skin [4]. The gene encoding for IL-31 is in the long region of chromosome 12 and encodes an mRNA string composed of 904 nucleotides. Various single-nucleotide polymorphisms (SNPs) have been suggested as possible risk factors for the development of itching diseases, but no conclusive result was found [5].

### 1.3. IL-31 Receptors

This cytokine, on the receptive end, presents a receptor complex composed of IL-31RA, a gp130-like receptor (GPL), and an oncostatin M receptor (OSMR) [6]. The receptor–ligand interaction leads to an activation of the STAT3 and STAT5 pathways, while the STAT1 pathway does not present a strong activation [6]. Shc- and SHP-2-adapter molecules are also involved in this signaling pathway, with an increment of MAPK activation [7]. This receptor is scarcely expressed on the cell surface, when in physiological conditions, and is upregulated by interferon γ (IFNγ) and toll-like receptor 2/toll-like receptor 1 agonists [8], while IL-31 acts as a positive-feedback stimulus on the expression of the receptor itself, especially when, firstly, secreted via the positive feedback of IL-33 [9]. The activation of this axis leads to several effects, such as the secretion of pro-inflammatory cytokines, tissue remodeling, and cell proliferation [9].

### 1.4. IL-31 Function

Although the pleiotropic role of IL-31 has been studied in several diseases, the most prominent role is represented by the pathogenesis of pruritus, especially in skin conditions such as atopic dermatitis (AD). Compared to histamine-mediated pruritus, IL-31 stimulates a later-onset pruritus [10]. Firstly, IL-31 has been studied in patients affected by atopic dermatitis, in which this cytokine is the major promoter of pruritus and scratching behavior. Among its activities in this disease, IL-31 also promotes epidermal-cells’ proliferation and thickening of the skin via skin remodeling, during the chronic Th-1-mediated phase of atopic dermatitis [9]. Stott et al. point out that IL-31 cannot be considered a Th-2 cytokine in the classical sense of the term, since it can be produced by another kind of T-helper cell in atopic conditions [11]. Indeed, its role in the chronic state of atopic dermatitis, in which Th1-mediate inflammation is involved, has been established [12]. This variety of expression depends on the microenvironment: the autocrine IL-4-expression stimulates Th1 clones to express IL-31 [13]. Moreover, other factors can modify the production of IL-31 by immune and non-immune cells, such as exposure to allergens, pathogens, and UV radiation [3]. Murdaca et al. reviewed a study about an in vivo allergic asthma model, which suggests that Th2 cytokines are the main triggers of IL-31RA expression and play a crucial role in Th2-mediated IL-31/IL-31RA connections [9]. IL-31 binds to its receptor complex (IL-31RA), which heterodimerizes with OSMRβ [14] and leads to the activation of three signaling pathways [9] (Figure 1):
(1)JAK/STAT pathway (Janus-activated kinase/signal transducer and activator of transcription),(2)PI3K/AKT (phosphatidylinositol 3′-kinase/protein kinase) pathway,(3)MAPK-JNK/p38 (mitogen-activated protein kinase—Janus kinase/p38) pathway.

**Figure 1 ijms-23-06507-f001:**
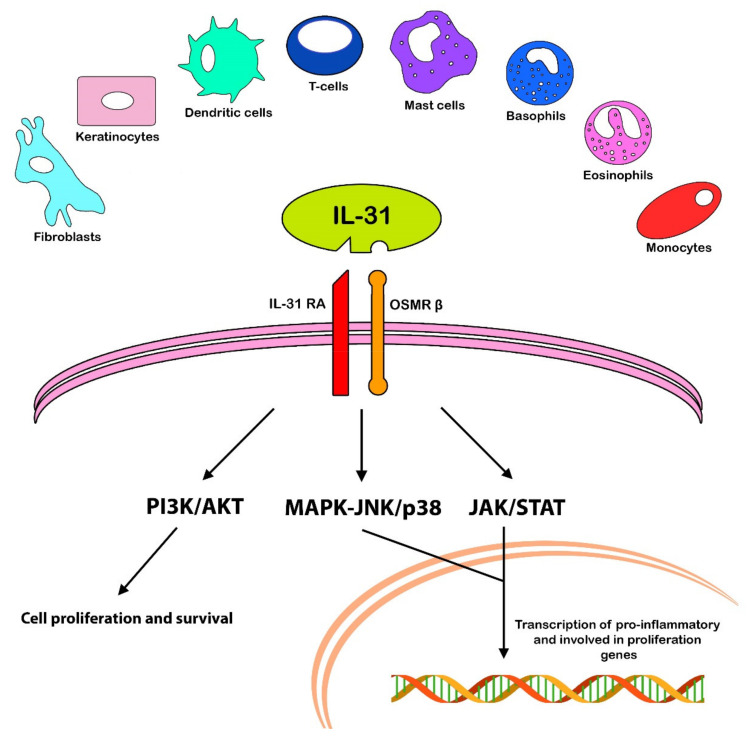
Representation of the main cells involved in the production of IL-31 and the three intracellular-signaling pathways, which are activated following the binding of IL-31 with its receptor.

Via phosphorylation of JAK1/2 and PI3K/Akt and the consequent activation of the STAT pathway (STAT3 and STAT5), IL-31 causes skin inflammation and proceeds to the dysregulation of the immunomodulatory proteins in atopic dermatitis [8]. STAT3 activation induces the expression of suppressor of cytokine signaling 3 (SOCS3), responsible for a negative-feedback mechanism: SOCS3 inhibits IL-31 signaling through inhibition of JAK activity. Via the binding to its receptor and the consequent signaling pathways, IL-31 could play a double role, acting as an early proinflammatory regulator and, subsequently, performing an anti-inflammatory activity to suppress the extent of type 2 inflammation [15]. Lastly, IL-31’s direct immunomodulatory properties have been documented in vitro: the stimulation of human-epidermal keratinocytes with IL-31 induced the expression of the thymus and activation-regulated chemokine (CCL17) as well as macrophage-derived chemokine (CCL22), suggesting that IL31 might be involved in the recruitment of polymorphonuclear cells, monocytes, and T cells to the skin [5].

## 2. Discussion

### 2.1. IL-31: Role in Skin Diseases

#### 2.1.1. Atopic Dermatitis

IL-31 is a pruritogenic cytokine, and, along with IL-33, is an alarmin involved in the inflammation axis IL-31/IL-33. Together they relate to the pathogenesis of atopic dermatitis (AD), in which eosinophil infiltration into the inner-dermal compartment represents a predominant pathological feature. IL-31 triggers the surface expression of the intercellular adhesion of molecule-1 (ICAM-1) on eosinophils and fibroblasts. In addition, the interaction between these two types of cells under IL-31 and IL-33 stimulation activates extracellular-signal-regulated kinase and induces pro-inflammatory cytokine IL-6 and other AD-related chemokines, such as CXCL1, CXCL10, CCL2, and CCL5. The findings suggest a crucial immunopathological role of IL-31 and IL-33 in AD, through the activation of eosinophils–fibroblasts interaction [16]. Since these cytokines are involved in skin inflammation, the authors hypothesize that IL-31-specific activation of dendritic cells may be part of a positive-feedback loop of inflammation [17]. On another note, even the polymorphisms of IL-31 haplotypes could lead to different manifestations of atopic eczema, suggesting that some subtypes of this cytokine could lead to better or worse clinical pictures [18]. As a cytokine first isolated from mice by Dillon et al. [1], IL-31 has been found to be one of the main protagonists of AD-induced inflammation and pruritus, two of the main features of this cutaneous condition. In inflammation, as one of the Th2-profile cytokines, it has been suggested that IL-4 mediates IL-31RA expression, and IL-31/IL-31RA interaction augments other chemokine production, such as CCL 17 and CCL 22, in bone-marrow-derived dendritic cells (BMDCs). BMDCs are one of the first lines of cells that interact with external stimuli in the host’s defense, with consequent Th2-oriented immune response [19]. Immunohistochemical staining for IL-31 and IL-31RA has found that IL-31 expression is increased in the inflammatory infiltrates from skin biopsies taken from subjects with AD, compared with controls [20]. Other studies found correlations between the serum IL-31 level and the serum IgE, eosinophil cationic protein (ECP), disease severity, and subject-itch intensity to certain degrees in AD patients [4,21,22], even though this was not confirmed by other publications, which found that IL-31 levels do not relate to disease severity [23,24]. Nonetheless, some works suggest a possible role of IL-31 as a consistently reliable marker of disease activity through serum measurements [25]. A major function of IL-31 in AD is the induction of pruritus in the skin. Some studies showed that IL-31 induces pro-inflammatory cytokines after staphylococcal-exotoxins’ exposition in human macrophages, preventing S.-aureus-induced up-regulation of mRNA encoding for AMPs human β-defensin 2 and −3 [26,27]. However, the signaling pathways of IL-31 in activated human macrophages remain unclear. This effect of interleukin-31 is supposed to happen via activation of STAT-1 and STAT-5 but not STAT-3, in human macrophages after up-regulation of the IL-31 receptor with staphylococcal exotoxins. Another pro-inflammatory effect played by IL-31 in activated human macrophages is the IL-12 suppression. This mechanism may be relevant in Th2 inflammatory responses, especially considering the overlap of Th1-oriented response (due to bacteria colonization) and baseline Th2 activation (typical of AD disease). The effects of IL-31 on AD have demonstrated that its effects could be modified by other conditions and agents. A study has found that reduced levels of vitamin D in the sera of these patients leads to poor production of LL-37, an antimicrobial molecule secreted by neutrophils, which correlates to AD-manifestation severity. LL-37 may systemically potentiate oncostatin M and IL-31 production in patients with AD, while vitamin D3 may do so only in non-affected people [28,29]. The role of IL-31 in the development of itch has been demonstrated in a line of DOCK8-deficient mice, in which the CD4+ T cells produce large amounts of IL-31. The induction of this cytokine, according to some findings, depends on the transcription factor of endothelial PAS domain protein 1 (EPAS1), and its deletion in CD4+ T cells abrogates skin-disease development in DOCK8-deficient mice [30]. The itching in AD patients is so relevant for the physiopathology of this disease that the related scratching behavior leads to an increase in pro-inflammatory mediators, including histamine, TSLP, IL-31, and substance P as well as changes in skin pH and trans-epidermal water loss (TEWL). Even the simplest method, nail clipping, leads to a decrease in pruritus and skin-barrier defects. In a study involving mice, immunofluorescence staining and Western blot results revealed that the antipruritic effect of nail clipping or the application of a plant-based topical treatment might be explained by the suppression of IL-31 [31]. This finding alone suggests the possible role of physical stimuli as triggering factors for the secretion of IL-31, although another interesting study suggested that even intense physical activity could lead to spikes in IL-31 production [32]. Several studies have suggested IL-31 as a potential target for future therapies for AD [33]. The main reason to treat patients with IL-31 antibodies is that the scratching behavior leads to the local thickening of the skin and the perpetration of an itching sensation. First-line therapies include phototherapy, antihistamines, and topical steroids. Although the effects of antihistamines do not trump other treatments, adding oral fexofenadine to the topical treatment for AD led to the conclusion that combined treatment significantly lowers serum IL-31 levels in patients [34]. Regarding UVB, IL-31 is upregulated in acute phases of sunburn, probably due to UVB and radical-oxygen species (ROS) activity. Chronic low doses of UVA-1 and UVB lower the levels of serum IL-31, thus, there is a double role of UVB, both triggering and dampening, in the production of IL-31 [35]. The resolution of AD lesions following phototherapy is accompanied by a significant reduction of mRNA expression of IL-5, IL-13, and IL-31, supporting current concepts that these cytokines play a crucial role in its pathogenesis and possibly represent targets for phototherapy [36]. On the other hand, hβD-1, IL-4, IL-10, and IL-31 are expressed in acute skin lesions in AD, and their levels correlate with each other [37]. The role that IL-31 plays, both in sunburns and AD-manifestation development and pruritus, suggests a common ground where UVB rays may play both a protective and pejorative effect. This consideration suggests that, as a therapy, UVB treatment should be administered with caution, to prevent patch development and avoid side effects that might subside if treatment protocols are not respected. Along with topical corticosteroids, other substances have proven useful in AD treatment. Some plant extracts act as diminishers of IL-31 production topically, such as resveratrol, rice components [38], and *P. densiflora* bark extract, all acting as IL-31 mRNA inhibitors [39]. Some of the most promising substances are coumarins from the fruit of *Cnidium monnieri* (TCFC), which possess evident biological activities based on their compounds. TCFC upregulated the filaggrin mRNA in the skin of rats, while it downregulated the levels of IL-1β, IL-4, IL-31, and TSLP mRNA [40]. Another aspect to consider is gut microbiota, since increasing studies have shown a close relationship between gut and skin health. In a mouse model, administration of *L. fermentum* KBL375 decreased the dermatitis score, skin thickness, and serum-immunoglobulin-E level in AD mice. This decrease in disease activity came along with decreased levels of IL-4, IL-5, IL-13, and IL-31, while anti-inflammatory cytokine IL-10 and transforming-growth factor-β increased, if the skin samples were treated with *L. fermentum* [41]. Although these studies are more frequently proposed in recent years, there is too little evidence to suggest final considerations.

#### 2.1.2. Psoriasis

Although not prominently studied in psoriasis (Pso), since the main actors in its pathogenesis are Th1 and Th17 interleukins, IL-31 has shown, over the years, a certain role in Pso itching, leading to the concept of “itchscriptome”. In a study using three groups (Pso, AD, controls), the levels of pro-inflammatory cytokines were assessed and, along with IL-17 and -23, common targets for current therapies in Pso as well asIL-31 resulted in elevated levels in the patients’ sera [42]. A study, focused on defining the difference between AD and Pso expression of genes, found that TRP vanilloid 2, TRP ankyrin 1, protease-activated receptor 2 (PAR2), protease-activated receptor 4 (PAR4), and IL-10 were increased only in atopic skin, while the expression of genes for TRP melastatin 8, TRP vanilloid 3, phospholipase C, and IL-36α/γ were elevated only in pruritic-psoriatic skin [42]. UVB therapy is a well-known therapy to treat psoriasis, especially the widely spread subtypes, such as chronic-plaque psoriasis and guttate psoriasis. These forms are the two mainly manifesting with skin pruritus and, in a 2013 study, NB-UVB exposures caused a significant decrease in IL-31 level. NB-UVB therapy had no substantial effect on the brain-derived neural factor (BDNF) involved in pruritus, regardless of the number of irradiations. Based on this study, IL-31 seems involved in the pathogenesis of psoriasis, while NB-UVB therapy could prevent the evolution of patches, by lowering the levels of this cytokine [43]. According to other studies, though, the morphological phenotype does not seem to be an important factor affecting the prevalence and characteristics of pruritus in psoriasis [44]. Another typical skin disease, in which itching plays a predominant role, is chronic urticaria (CU). A 2020 study measured levels of IL-31 in patients affected by both Pso and CU, showing that higher concentrations of this interleukin are significantly related to the coexistence of these two conditions. The authors of this study did not suggest the idea that the levels could be higher just due to the presence of CU, especially since IL-31 is a Th2-oriented cytokine, but the results suggest a common therapeutic ground for patients affected by both conditions [45].

#### 2.1.3. Chronic itch

Chronic itch represents a non-secondary issue in modern-day practice. Many patients are diagnosed with a chronic itch of unknown origin, mostly suggesting systemic conditions such as kidney failure, skin xerosis, hematological diseases, liver failure, and diabetes, just to name a few. If the pruritus remains untreated in time, it can lead to nonspecific conditions such as prurigo nodularis. Before reaching visible-cutaneous manifestations, a study of a common cause of itching could prove useful. Some reports have found differences in serum-uric acid, ALT, and AST between patients and controls, but this elevation was accompanied by higher IL-31 levels [46]. Experiments conducted on mice show that IL-31 upregulates IL-31RA expression in the root ganglia of neuron-cell bodies, and cutaneous-injected IL-31-induced itching is enhanced by DRG IL-31RA expression in mice [47]. These findings suggest a vicious cycle involving IL-31 and the perpetration of the itching stimulus also found in humans. Patients with AD experience increased sensitivity to minimal stimuli, which induce sustained itch [48]. In this regard, IL-31 has been found to induce a transcriptional program in sensory neurons, which leads to nerve elongation and branching. In another study, itch-sensing skin has been found to exhibit free endings with extensive axonal branching in the superficial epidermis and large receptive field [49]. These data suggest that further studies, in the future, might find common pathogenetic ground in conditions such as chronic itch and the possible development of other pain-related conditions such as fibromyalgia, for which the pain threshold is lower than in healthy individuals. Several questions remain unsolved, though, on the mechanism of the action of IL-31 in chronic itch. IL-31-receptor immunoreactivity was observed in the epidermis and primary-sensory neurons. IL-31 increased the production of leukotriene LTB4 in mouse keratinocytes in which IL-31 receptor mRNA is expressed, suggesting that IL-31 elicits itch not only through direct action on primary sensory neurons but also by inducing LTB4 production in keratinocytes [50]. Since many diseases commonly cause itching, most of the studies proposed to investigate pruritus as a clinical condition linked mostly to atopic disease and allergic conditions in general. Although Th2-oriented skin conditions do not manifest solely through itching, pruritus can often appear before any skin manifestation. Figure 2 represents a summary of such mechanisms.

### 2.2. IL-31: Role in Allergic Diseases

#### 2.2.1. Allergic Respiratory Diseases

In the last years, the interest in the study of IL-31 has grown concerning, in addition to skin conditions, Th2-mediated airways diseases, especially allergic rhinitis and asthma. Indeed, IL-31 is produced by Th2 clones, eosinophils, and mast cells, all cytotypes involved in the pathogenesis of the allergic inflammation of the airways with the help of IL-4, a pivotal cytokine in allergic inflammation, which promotes the production of IL-31 through an autocrine mechanism involving Th-2 cells [11]. In fact, according to a study [51] conducted on animal models to determine the role of IL-4 and IL-13 in modulating the expression of IL-31RA, it was found that macrophages stimulated with IL-4 and IL-13 increased IL-31RA, and cells treated with antibodies against IL-4R blocked the following signaling pathway involving STAT6, which decreased IL-31RA expression.

Regarding asthma, some authors tried to detect IL-31RA on pulmonary epithelial cells, finding that IL-31RA was expressed at clearly visible levels using confocal microscopy in cultures of pediatric-bronchial-epithelial cells [52]. Another study found a positive correlation between serum IL-31 levels and asthma-disease severeness, along with the expression of other Th2 cytokines, such as IL-5, IL-13, and TSLP [53], while the study group of Lei et al. [54] suggested that IL-31 could be a useful indicator for asthma activity. They observed that IL-31 mRNA levels in peripheral-blood mononuclear cells (PBMCs) were higher in allergic asthmatic. Similarly, Moaaz et al. [55] obtained the same results by measuring IL-31 mRNA levels in PBMCs. They suggested that the activation of the MAPK pathway could be crucial in bronchial inflammation and the release of epithelial-growth factor (EGF), vascular-endothelial-growth factor (VEGF), and CCL2, all of which belong to pro-inflammatory mediators and remodeling factors. They also found a statistically significant increase in IL-31 in atopic patients, when compared to non-atopic patients, and a positive correlation between IL-31 and total IgE levels, although a previous study [56], which assessed the role of IL-31 mutations in higher risk of asthma development, did not confirm this hypothesis. Another strain of research regarding bronchial epithelium of children with asthma found some correlation with decreased levels of 25-OH vitamin D and higher levels of IL-31 [57], but further research should be carried out to assess the possible therapeutic options [58,59]. Lastly, Ulambayar et al. [60], in 2019, suggested that in elderly patients, the decrease in IL-31 and IL-33 levels were related to different clinical phenotypes and severity of asthma, perhaps due to lower levels of Th-2 inflammation.

On the other hand, the protective role of IL-31 against Th2-lung inflammation has been demonstrated in several studies. Neuper et al. showed that overexpressing-IL-31 mice had lower leukocyte infiltration in the lungs, reduced mucus production, and epithelial thickening [61], while Huang et al. demonstrated that a lack of IL-31/IL-31RA-signaling pathways resulted in exacerbated inflammation and increased IgE, IL-4, and IL-13 levels [15]. Concordantly to these results, Perrigoue et al., earlier, suggested the possible role of IL-31/IL-31RA signaling in self-limiting type 2 inflammation [62].

Regarding allergic rhinitis (AR), both Okano et al. [63] and Baumann et al. [64] found that allergen-stimulated PBMCs produce IL-31 and correlate to the severity of the disease and its symptomatic manifestations, thus suggesting that IL-31 may increase inflammation in the nasal epithelium through the release of mediators (CCL17, CCL22, and CCL1), which recruit inflammatory cells, as seen in AD.

Another study [65] found higher levels of IL-31 and IL-31RA in mucosal samples of AR patients, correlated with the gene expression of MUC5AC (considered to be the major mucin in the airway), providing evidence that IL-31 also plays a role in mucus hypersecretion in AR patients. This finding was corroborated in a cohort of pediatric patients [66], showing higher IL-31 levels in the serum, mucosal samples, and nasal lavage of patients with AR, compared with controls, especially in those with concomitant asthma, along with IL-4, IL-5, IL-13 and MUC5AC. This evidence suggests the role of IL-31 in enhancing Th2 response and eosinophils activation in AR. In fact, in mice deficient in IL-31RA, lower responses to injected nasal allergens were noted [67], compared to wild-type mice. Overall, other studies demonstrate that IL-31 correlates to asthma and allergic rhinitis [68], but may have many limitations, especially regarding the number of patients recruited, and some are discordant with each other. To elucidate the role of this interleukin in respiratory-allergic diseases, we believe further in vivo and in vitro studies are needed to outline mechanisms of action as reproducible and standardized, as possible.

#### 2.2.2. Urticaria and Allergic-Contact Dermatitis

Regarding allergic-skin disorders, the role of IL-31 has also been studied in allergic-contact dermatitis (ACD) and chronic-spontaneous urticaria (CSU). These diseases feature intense itchy cutaneous manifestations that severely impact the quality of life of affected patients and sometimes can prove refractory to several therapeutic options.

Concerning CSU, the main actors are basophils and mast cells, via the interaction of IgE with its high-affinity receptor (FcεRI). Associations between CSU and autoimmune diseases have also been described, since in about one-third of CSU patients, antibodies against IgE or FcεRI were found that directly stimulate mast cells and basophils [45]. Beyond mast cells and basophils, other immune cells play a role in CSU patients, such as IL-10 and IFN-γ-secreting T cells [69]. Studies regarding IL-31 and urticaria were mainly concerning disease severity and therapy exploitations, as basophils have been found to produce pro-inflammatory cytokines (IL-4 and IL-13), after the interaction of IL-31/IL-31R [70]. The stage of the disease and the severity of the itching have been correlated to serum IL-31 levels in CSU patients. Some authors [45,71,72] found that patients with CSU had elevated serum levels of IL-31, when compared with healthy controls, while others did not find any correlations [73], thus suggesting that IL-31 does not act as the main actor in wheal number or frequency. On the other hand, regarding pruritus, Lin et al. [72] found a positive correlation with high serum IL-31 levels. Indeed, in agreement with previous evidence, they did not find a correlation between IL-31 levels and urticaria activity measured via the urticaria-activity score after seven days (UAS7), while serum IL-31 levels in CSU patients with severe pruritus were higher than in the mild group. As for the therapy, a reduction in serum IL-31 levels and fewer IL-10-, IL-31-, and IFN-γ-secreting T lymphocytes were found in CSU patients treated with omalizumab, respectively, by Altrichter et al. [73] and Rauber et al. [74], but no IL-31 blockade for therapeutic purposes has been proposed so far.

Regarding ACD, Guarneri et al. [13] were the first to evaluate a possible systemic role of IL-31 in this cutaneous disease, as the main mediator of pruritus, regardless of the different allergens to which patients were sensitized. In 2006 [75] and 2018 studies [76] levels of IL-31 were found to be higher in skin samples of patients affected by ACD, suggesting a critical role in itch stimulation, but not in pathogenetic processes such as induction of inflammation and hapten-specific T-cell activation. As for other skin diseases, IL-31 plays some role in itch development, but studies are too scarce to draw final conclusions. The results collected through our research on allergic-skin diseases are still few and very discordant. However, interest in this topic has increased in recent years, registering two peaks of interest, in 2017 and 2020. It can be said that, as the advent of newer biological drugs are developed more and more frequently, new trials regarding these allergic conditions could be carried out in the future. A summary of allergic diseases and the role of IL-31 is pictured in Figure 3.

### 2.3. IL-31 in Hematological Diseases

The role of IL-31 in hematology has been mainly assessed as a pruritogenic intermediate in tumors. Itching is experienced by 50% of patients with Philadelphia chromosome-negative-myeloproliferative disorders (Ph-MPDs) [77,78], 66–88% of patients with advanced Cutaneous T-cell Lymphoma (CTCL) [79], and about 83% of patients with mastocytosis [80]. Itching can be the first sign of an oncological disease, appearing as a paraneoplastic syndrome related to mediators released by cancer cells, and research has focused its lens on cytokines as possible itch mediators. Among hematological diseases, we have found that the role of IL-31 was mainly studied in CTLC. Levels of IL-31 in the sera of patients suffering from mycosis fungoides (MF) and Sézary syndrome (SS), the most frequent forms of CTCL, were found to be higher than in healthy controls, and higher levels of IL-31 were related to disease severity [81]. In another study, Miyagaki et al. evaluated the role of CCL18, already used as a severity index in AD and bullous pemphigus, in patients suffering from CTCL; its levels, along with IL-31, are related to a worse prognosis [82], while Singer et al. investigated the correlation between levels of IL-31 and itching severity in patients with CTCL, confirming an increase in IL-31mRNA in the PBMCs in patients who complained of itching [83]. Conversely, Malek et al. [84] did not find this correlation in just any patient with CTCL, but this can be explained by the examination of patients in different disease stages, thus not allowing comparable results. Overall, results indicate that IL-31 levels relate to itch severity but not necessarily to disease stage [85]. Perhaps, different stages of the same pathological entities could explain the different results [86].

Moving to mastocytosis, Hartmann et al. [87] noticed increased IL-31 levels in the sera of patients with mastocytosis that were related to the severity of the disease, serum-tryptase levels, and the percentage of bone marrow infiltration. They demonstrated that mast cells are the main source of IL-31 in mastocytosis in both the skin and bone marrow, while another study correlated itch intensity and sera levels of this cytokine [80], although some other factors could be involved in the pruritus, such as histamine or a possible keratinocyte-derived mediator, which subsequently activates unmyelinated-itch fibers [88].

In conclusion, in the hematological field, IL-31 has been studied for its role in itching. Some authors related IL-31 levels to itching severity and the stage of cancer. Although not resolutive, treatments aimed at decreasing the levels of IL-31 could represent, in the future, possible means to better the quality of life of patients affected by unbearable itching.

### 2.4. IL-31 in Oncology

In recent years, the study of cytokines in oncology has undergone a notable increase in interest, due to the possibility of evaluating new factors predicting the onset and progression of a neoplasm, and the chance of identifying new therapeutic targets.

Since IL-31 has been studied for the first time in epithelial cells, the first tumor of interest was lung cancer [89]. In the lung, the main targets of IL-31 are bronchial and alveolar epithelial cells, pulmonary fibroblasts, and macrophages, which can promote the expression of EGF and VEGF, with levels that were measured via bronchoalveolar-lavage-fluid (BALF) samples. On a side note, they noticed that IL-31 levels correlated to disease-progression speed.

In oncologic patients, IL-31 is often produced by damaged and inflamed tissues, especially by mast cells. Zeng et al. [90] found higher levels of IL-31 in the sera of patients with endometrial cancer than in healthy controls, but the relationship between this finding and the clinical evidence was not clear. They hypothesized that IL-31 could be produced by mast cells after IL-33 stimulation. Later, the same authors [91] conducted another study about IL-31 and IL-33 in endometrial-cancer patients: they found higher expression of IL-31, IL-31R, IL-33, and ST2 in malignant tissue than in the healthy control group, and they observed that a stronger expression of IL-31 and IL-33, as well as their receptors, were associated with a worse stage and a worse prognosis.

More recently, different authors have reported the antiproliferative and antiangiogenic role of IL-31 in cancer cells. Kan et al. [92] demonstrated that IL-31 expression inhibits tumor growth only in immunocompetent mice but has no effect on tumor growth in immunodeficient NOD/SCID (nonobese-diabetic/severe-combined immunodeficiency) mice. They suggested that the antitumor activity of IL-31 affects cytotoxic T lymphocytes (CTLs) via decreased levels of IL-10 and an increase in total CTLs. Switching to ovarian cancer, the JNK-STAT pathway in tumor invasion and metastasis is activated by IL-31 in the overexpressing, tumoral cells (IL-31RA+), with a possible correlation with different clinical presentation and prognosis [93]. As with ovarian cancer, as well as with breast cancer, IL-31 might play a role in the metastasis and migration of tumoral cells [94]. Soon, new therapeutic strategies could be developed, in consideration of the expression of IL-31RA by tumors and their responsiveness to anti-IL-31 drugs. In recent years, studies on IL-31 in oncology have also focused on the variability of expression of the IL-31 gene, the identification of SNPs, and the correlation with clinical characteristics and tumor stages.

Before consistent evidence can be produced about anti-IL-31 therapies in solid tumors, the assessment of different nucleotide sequences could represent a prognostic factor, to be taken into consideration for the total survivability, such as rs7977932 [95], which represents a predisposing factor. Another study revealed the probable involvement of the same IL-31 polymorphism in endometrial-cancer susceptibility, thus representing a novel genetic marker [96]. Moreover, SNP rs4758680 could be an independent risk factor for the decreased recurrence-free survival of patients affected by bladder cancer [97]. All these findings suggest that SNPs of IL-31 could have a role in the diagnosis or prognosis of several epithelium-derived solid tumors, especially when in combination with other tumor-related markers. Figure 4 resumes the evidence regarding the role of IL-31 in solid tumors.

### 2.5. IL-31 in Infectious Diseases

Another field in which the recent studies delved in to research the role of IL-31 has been infectious diseases. As a cytokine that acts upon a certain number of cells’ subpopulations, IL-31 was researched in systemic conditions such as sepsis, respiratory diseases, and HBV, beyond the well-known pruritogenic role that it plays in itching diseases such as scabies [98]. The most relevant results regarded sepsis. The sepsis-related syndromes can be identified and diagnosed with other means, but interleukin-31 seems to predict their severity. IL-31 was able to reduce the mortality rate of lipopolysaccharide (LPS)-induced sepsis, by a reduction in the inflammatory cytokines and inhibition of IL-1β production in LPS-induced sepsis. The most interesting role is the inhibition of the expression of NLRP3 at the transcriptional level. NLRP3 belongs to the inflammasome structure, meaning the oligopeptide-multiprotein system, which regulates another form of programmed cell death, called pyroptosis. This process leads to a cascade of activation of different caspases, which ultimately causes the destruction of the cell by the formation of multiple surface pores and the consequent release of proinflammatory cytokines, in particular IL-1beta, IL-18, and Gasdermin-D. In human cells, neutralization via antibodies against IL-31 and its receptor enhanced NLRP3 expression and IL-1β activation, suggesting that the inflammasome plays a role in sepsis development. At the same time, IL-31 down-regulated NLRP1 expression with lesser expression of other inflammatory cytokines, such as TNF-α [99]. Thus, arguably, systemic-inflammatory-response syndrome (SIRS) and sepsis could be differentiated by the measurement of IL-31, IL-1ß, and NLRP3. Lower levels of IL-31 are characteristic of sepsis, when compared to SIRS, and even lower levels in the case of septic shock, showing sensitivity up to 80% [100]. These findings suggest that, by studying further the role of IL-31 in sepsis, bioproducts that could lead to the blockade of IL-31 could be used, in the future, to control the symptoms and complications caused by septic-shock syndromes, in which patients’ survival depends largely on the control of the heavy-inflammatory status. Regarding HBV, just two articles were identified, which shows that this condition requires further investigation, considering that liver damage is scholarly linked to generalized pruritus, and IL-31 is a pruritogenic cytokine. Nonetheless, a study investigating the association between TGF-β1/IL-31 and stages of chronic HBV infection showed that TGF-β1 and IL-31 are linked to the progression from chronic hepatitis to cirrhosis and correlate with the severity of cirrhosis. It can be suggested that the TGF-β1/IL-31 pathway could be related to the pathogenesis of liver fibrosis in chronic HBV infection [101], and decreased levels of the above-mentioned markers are typical of healing injuries of hepatic scars. Moreover, levels of TGF-β1 and IL-31 were linked to survivability of these patients [102]. These findings suggest that IL-31 may play other roles in other axes involving systemic conditions. Finally, regarding the evident role of Th2-oriented response in parasitic infections, IL-31 has been studied also in gastrointestinal colonization and pulmonary infections. There is a report on the regulatory role of IL-31 interacting with its receptor in the intestine, after gastrointestinal infection by a parasite (*Trichuris muris*), which triggers a Th2 response and the production of IL-4 and IL-13. In response to *Trichuris* infection, KO mice for IL-31R-alpha had a stronger response in their lymph nodes, when compared to controls, with goblet cells’ hyperplasia accompanied by a higher rate of mucosal secretion in the intestine, suggesting a stronger response to the infection [103]. In the respiratory tract, levels of IL-31 were assessed while in the presence of an *Echinococcus multilocularis* infection. In this case, patients were divided into three branches (cured, stable, and progressive) to differentiate their immunological response, based on disease stage. The authors of the study noticed that the IL-31/IL-33 axis was depressed in the case of all disease stages, while regulatory molecules, such as IL-27 and eosinophil-granulocyte-attracting eotaxins, were enhanced in case of progressive disease, thus suggesting that some markers can be identified to monitor the disease activity in the case of parasitic infections [104].

### 2.6. Miscellaneous

From the researched articles, it was not possible to group several studies into macro-areas in such a way as to obtain significant data. The trend of these articles shows that the interest in Il-31 has significantly increased in the last five years, in various fields of medicine, such as rheumatological diseases, neurological diseases, and other itchy-skin conditions. Hopefully, in the future, further studies will be carried out in this direction to confront and assess a more defined role of this cytokine in other pathologies.

## 3. IL-31: Target for Current and Future Therapies

Finally, with the advent of biological treatments, we reached a turning point, as suggested by Gangemi et al. [10]: the use of monoclonal antibodies against the mediators involved in the pathogenesis of pruritus in skin diseases is very promising in the control of pruritic symptoms. Considering that the itch represents a primordial-defense mechanism, which can be restrained by the will of the patient only to a given extent, the control over this symptom is one of the main targets of current therapies aimed at bettering the quality of life.

In a control study dealing with mice specimens, the scratching behavior induced by IL-31 was inhibited if the specimen was administered with an antibody blocking the receptor of IL-31. Such itching was inhibited more than when compared to the administration of antihistamines or naloxone. The anti-IL-31-receptor antibody reduced swelling and dermatitis score (a possible relation to SCORAD in men) in a chronic pruritus-inducing AD-like model [105]. To treat the itching, independently linked to a specific condition, several AD-like models have been previously developed, mostly in canine specimens. Lokivetmab, as one of the first and most-studied monoclonal antibodies, seems the most successful drug developed so far. Although first-line treatments, such as corticosteroids and oclacitinib, have produced remarkable results in itch control [106], in a 2018 study, pruritus improvement was achieved in almost 90% of cases after administration of lokivetmab, without any change in the outcome if the treatment was administered to different specimens (i.e., diseases’ chronicity or age of onset) [107]. As a comparison, though, it must be emphasized that, while specimens that did not respond to oclacitinib also did not respond to lokivetmab, this monoclonal antibody targets just one itch mediator specifically, namely IL-31. Oclacitinib, as a JNK inhibitor, on the other hand, can produce other side effects if not tolerated by the individual, such as cutaneous lumps, serum-lipid increase, and gastrointestinal-tract symptoms [107], with a lower safety profile when compared to the anti-IL-31 molecule.

These types of studies led to the development of a specific molecule, nemolizumab, for human patients. In a 24-week study, nemolizumab was administered every 4 weeks in adults with moderate-to-severe AD, improving the cutaneous conditions of inflammation and pruritus in patients, when compared to a placebo, keeping a high safety profile [108]; according to another study, similar results were achieved in a 64-week period [109]. Although studied just in AD-like models in humans, nemolizumab could also prove useful in other chronic itching conditions, such as chronic kidney failure. In a 2020 work, uremic patients were enrolled in a study model to assess their levels of circulating IL-31: in uremic patients, the levels of this interleukin were significantly higher compared to the control group (*p* = 0.0001) [110]. In the same study, high levels of IL-31 were found both in uremic patients and cardiovascular patients, suggesting a higher chance of developing heart and blood-vessel disease in uremic-oriented conditions [110]. Patients that present higher levels of IL-31, and, thus, itching, could represent a higher-risk population, suggesting that itching might be a symptom correlated with disease severity. Moreover, treating uremic patients with anti-IL-31 antibodies could not only improve the itching but also represent a safer treatment to be administered to more fragile individuals, who could not bear chronic doses of oral or topical steroids and the already-cited JNK inhibitors. As a possible countermeasure for another itchy disease, meaning psoriasis, an inhibitor of the Janus kinase pathway, such as tofacitinib, has been proposed. In a mouse model with psoriasis-like imiquimod-induced dermatitis, tofacitinib led to an amelioration of the skin lesions, according to this theory. Again, as a possible common-therapeutic ground for patients affected by hybrid forms of generalized eczematous dermatitis (AD/Pso), JAK inhibitors could represent a future solution for the issues of these patients [111]; though, as mentioned above, side effects could represent a limit to such oral therapies, especially in consideration of the frequent association of psoriasis and metabolic syndrome. Finally, another promising molecule is represented by CIM331, which is a humanized IL-31RA monoclonal antibody, binding to IL- to inhibit subsequent IL-31 signaling, with improvement of pruritus and use of topical corticosteroids [112]. These two biological drugs, although still in the trial phase before being put on the market, could be useful someday for atopic patients, representing the need for safer and long-lasting therapies.

Concerning the therapeutic implications of IL-31 in CTCL, Cedeno-Laurent et al. [113] showed a decrease in IL-31 expression for PBMCs treated in vitro with vorinostat, a histone-deacetylase inhibitor (HDACi), or with dexamethasone in patients with stage IV CTCL. They also analyzed blood samples of patients with stage IV CTCL before and after treatment with romidepsin, another HDACi, demonstrating a reduction in IL-31 levels and a consequent reduction in itching. Lastly, they demonstrated that IL-31 in CTCL patients is produced by specific T-cells (CD4+/CD26-), which express the skin-homing chemokine-receptor type-4 (CCR4). Therapy with mogamulizumab, a monoclonal-antibody anti-CCR4, promotes a reduction in cancer-cell populations, followed by a decrease in IL-31 levels and itching. These clinical trials suggest that also targeting IL-31 in hematologic conditions could prove useful, not only for itch control but also regarding disease progression.

## 4. Conclusions

The studies collected deal with different pathologies, among which AD stands out: as chronic dermatitis is characterized by severe itching and involves a conspicuous share of the population, IL-31 has been studied and isolated here for the first time. As a cytokine mainly associated with itching, IL-31 was subsequently researched in other pathologies, mainly associated with hematology/oncology and Th2-immune-system responses, thus showing several cellular targets and mediators linked to IL-31 (Figure 5). As a marker, it has proven the potential of being used, in the future, to highlight acute phases of the disease or as a possible prognostic factor. As a target, the blockade of IL-31, via the molecule itself or its receptor, leads to the amelioration of itching, at the very least, and at best is a partial resolution for the condition manifestations. Arguably, new therapeutic approaches will be integrated with already-successful biological drugs, and also with anti-IL-31 antibodies, to target the whole spectrum of biological products that lead to the previously mentioned conditions. In fact, IL-31 has been targeted already with great success in animal models of AD, and, as in the case of the canine specimen, it is an already an available treatment. Arguably, the same type of Mab, such as nemolizumab, will be available for human patients as well, in the near future. The studies so far show that common topical therapies target the production of IL-31, although at the time of their discovery such effects could not be predicted (plant extracts, corticosteroids, antihistamines). IL-31 targeting has proven to be a safe and convenient method to treat itching, and it also has shown some potential in treating oncological diseases and a plethora of inflammatory conditions that could lead to organ failures, such as hepatitis and sepsis. Since its discovery, other authors have researched a possible causative role of this new cytokine in other conditions, such as other itchy dermatoses and rheumatic diseases, but, to this day, the number of studies is too low to draw any conclusion. The highlighted studies so far are too few and scattered in time to predict in which way future research will draw its attention, but, certainly, biological drugs will represent the safest route to treat several conditions, mostly skin-related, with open possibilities to treat even more concerning diseases (Table 1).

## Figures and Tables

**Figure 2 ijms-23-06507-f002:**
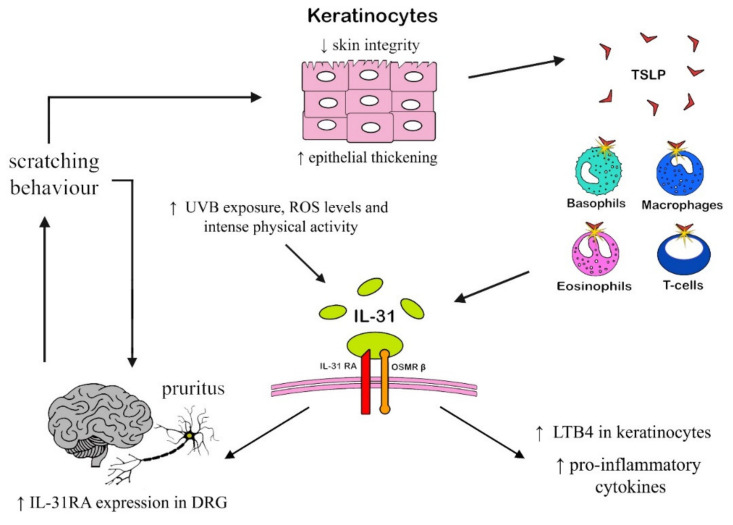
Main mechanisms involved in the onset of itching behavior in dermatological diseases (ROS = radical-oxygen species; DRG = dorsal-root ganglion).

**Figure 3 ijms-23-06507-f003:**
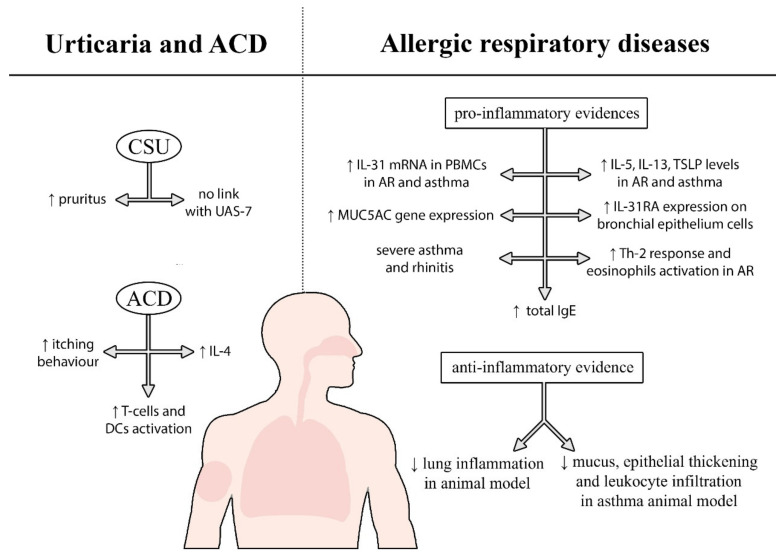
The main effects exerted by IL-31 in allergic respiratory and dermatological diseases (ACD = allergic contact dermatitis; CSU = chronic spontaneous urticaria; PBMCs = peripheral blood mononuclear cells; DCs = dendritic cells; AR = allergic rhinitis).

**Figure 4 ijms-23-06507-f004:**
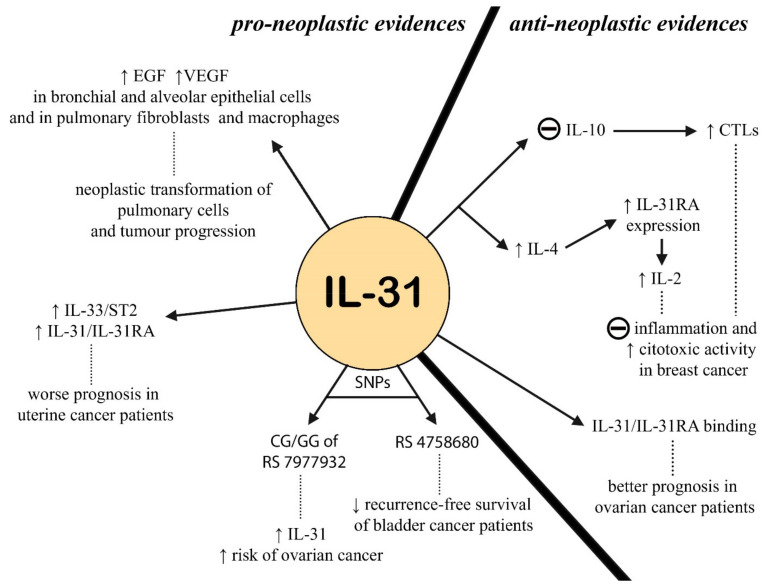
Role of IL-31 in oncological diseases: on one side is the mechanisms that could explain a pro-neoplastic action, on the other is the evidence that suggests an anti-neoplastic role (EGF = epithelial growth factor; VEGF = vascular endothelial growth factor; SNPs = single nucleotide polymorphisms; CTLs = cytotoxic T lymphocytes).

**Figure 5 ijms-23-06507-f005:**
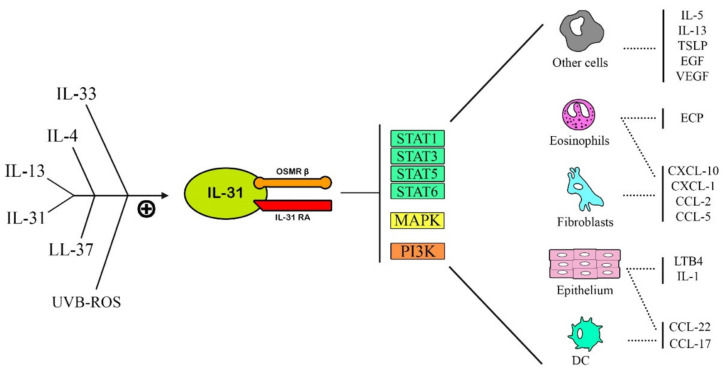
On the left, the major cytokines that stimulate the IL-31/IL-31R pathways are represented; on the right side of the figure, there are the effects of IL-31/IL-31R axis on the main target cells (DC = dendritic cell; ECP = eosinophils-cationic protein; EGF = epithelial-growth factor; VEGF = vascular-endothelial-growth factor).

**Table 1 ijms-23-06507-t001:** Summarizes the main cellular targets, the mediators involved, and the effects of IL-31 in the pathologies treated in the main text. AD, atopic dermatitis. AR, allergic rhinitis. CCL1,2,5,17,22 C-C Motif Chemokine Ligand 1,2,5,17,22. CTCL, Cutaneous T-cell lymphoma. CTLs, Cytotoxic T lymphocytes. CXCL1, 10, C-X-C Motif Chemokine Ligand 1,10. ECP, Eosinophil cationic protein. EGF, epithelial growth factor. HBV, hepatitis B virus. IL-1,4,5,6,10,13,18,31,33,36, interleukin 1,4,5,6,10,13,18,31,33,36. JNK-STAT, Janus kinase/signal transducer and activator of transcription. LTB4, leukotriene B4. MAPK, mitogen-activated protein kinase. MUC5AC, Mucin 5AC. NECs, nasal epithelial cells. NLRP1,3, NOD-, LRR-, and pyrin-domain-containing protein 1,3. PBMCs, peripheral-blood mononuclear cells. PSO, psoriasis. STAT-1,5, signal transducer and activator of transcription 1,5. TGF-β1, transforming growth factor beta 1. TNF-alpha, tumor necrosis factor-alpha. TRP, transient receptor potential. TSLP, thymic stromal lymphopoietin. VEGF, vascular-endothelial-growth factor.

Disease	Author/Citation	Target Cells	Mediators/Pathway	Effects
AD	Wong et al. [16]	Eosinophils—fibroblasts interaction	IL-33, IL-6, CXCL1, CXCL10, CCL2, CCL5	Skin inflammation
	Kasraie et al. [26]	macrophages	STAT-1 and 5	Itch
	Tang et al. [31]	keratinocytes	Histamine, TSLP, substance P	Itch, changes in skin pH, trans-epidermal water loss
PSO	Nattkemper et al. [42]	-	TRP melastatin 8, TRP vanilloid 3, phospholipase C, IL-36α/γ	Itch
CHRONIC ITCH	Arai et al. [47]	Dorsal-root ganglia	-	Perpetration of the itching stimulus
	Andoh et al. [50]	Mouse keratinocytes	LTB4	-
ASTHMA	Moaaz et al. [55]	Bronchial epithelium	MAPK, EGF, VEGF, CCL2	Bronchial inflammation
	Neuper et al. [61]	Bronchial epithelium of IL-31RA-/-mice	-	Lower-leukocyte infiltration, reduced mucus and epithelial thickening
AR	Baumann et al. [64]	Nasal epithelium	CCL17, CCL22, CCL1	Inflammation in the nasal epithelium
	Liu et al. [66]	NECs PBMCs	MUC5AC, ECP IL-4, IL-5, IL-13	Enhanced Th2 response and eosinophils activation
CTCL	Singer et al. [83]	PBMCs	-	Itch
LUNG CANCER	Naumnik et al. [89]	Bronchial and alveolar epithelial cells, pulmonary fibroblasts, and macrophages	EGF, VEGF	Increased cancer progression
MURINE BREAST CANCER	Kan et al. [92]	CTLs	Decreased levels of IL-10	Increase of total CTLs, antitumor activity
OVARIAN and BREAST CANCER	Wang et al. [93] He et al. [94]	- -	JNK-STAT -	Tumor invasion and metastasis
SEPSIS	Watany et al. [100]	-	Down-regulation of NLRP3, NLRP1	Pyroptosis, release of IL-1beta, IL-18 and Gasdermin-D, decrease in TNFalpha
HBV	Ming et al., Yu et al., [101,102]	-	TGF- β1	Liver fibrosis, progression to cirrhosis

## Data Availability

Not applicable.

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
