# Peer review of "IL-31: State of the Art for an Inflammation-Oriented Interleukin"

_ijms, 2022, doi:10.3390/ijms23126507_

Round 1

Reviewer 1 Report

The review article is very well written with a lot of figures that render the readability quite good. I have just a nimor comment in including at least one additional table summarizing findings from the main studies to be included in the conclusion remarks.

Reviewer 2 Report

In their review, Borgia and colleagues detailed the involvement of IL-31 in inflammation. Overall, the review is interesting, well organized, and the references are up-to-date and comprehensive. 

However, the following issues should be addressed:

1/ In my opinion, the "search strategy" section, Tables 1 and 2, and Graphs 1 and 2 are not very useful and do not help clarify the text. They should be deleted or added to the supplementary information. 

2/ Enthusiasm for the review is diminished by the inconsistent quality of the writing. For example, in paragraph 3.2, it seems to be a succession of descriptions of works. Thus, efforts to synthesize should be made in each paragraph and reduce the length of the manuscript. Sometimes there is too much detailed information that loses the reader. Altogether, these different points make the reader get lost quickly without knowing where the authors are going with their main message. 

3/ The figures contribute to a better understanding of the review. However, their quality should be improved. It seems that these figures were extracted from the slides of an oral presentation. In addition, the colors for IL-31 and the IL-31 receptor should be homogenized and the colored backgrounds should be removed. 

In conclusion, such a review is interesting to get a recent and comprehensive overview about IL-31 and its role in various diseases. However, it requires significant effort to obtain clear text and better quality figures.

Round 2

Reviewer 2 Report

The author did a great job, and significantly improved the quality of the manuscript.

I detected minor points:

Line 52,  "while the IL-31 acts" should be replaced by "while IL-31 acts"

Line 265, "coltures of pediatric..." should be replaced by "cultures of pediatric..."

Line 349, "Figure 3. The the main ..." should be replaced by ""Figure 3. The main ..."

Line 511, "I.E. diseases cronicity" should be replaced by "I.E. diseases chronicity"